# Microbial Landscape and Antibiotic Susceptibility Dynamics of Skin and Soft Tissue Infections in Kazakhstan 2018–2020

**DOI:** 10.3390/antibiotics11050659

**Published:** 2022-05-13

**Authors:** Sholpan S. Kaliyeva, Alyona V. Lavrinenko, Yerbol Tishkambayev, Gulzira Zhussupova, Aissulu Issabekova, Dinara Begesheva, Natalya Simokhina

**Affiliations:** 1Department of Clinical Pharmacology and Evidence-Based Medicine, NCJSC Karaganda Medical University, Karaganda 100000, Kazakhstan; s-kalieva@qmu.kz (S.S.K.); dr-minakova@mail.ru (N.S.); 2Scientific Research Laboratory, NCJSC Karaganda Medical University, Karaganda 100000, Kazakhstan; lavrinenko.alena@gmail.com; 3Department of Surgery, NCJSC Karaganda Medical University, Karaganda 100000, Kazakhstan; tishkambaev@qmu.kz; 4Salidat Kairbekova National Research Center for Health Development, Nur-Sultan 010000, Kazakhstan; 5National Center for Rational Use of Medicines, Nur-Sultan 010000, Kazakhstan; begesheva_de@mail.ru

**Keywords:** antibiotic resistance, antibiotic susceptibility, antibiotics, soft tissue infection, skin and soft tissue infection

## Abstract

Skin and soft tissue inflammatory diseases of bacterial origin occupy a significant part of hospitalizations to emergency departments. One of the most common causes of sepsis is soft tissue infection, which accounts for about a quarter of all nosocomial infections. The aim of this study was to determine the differences in microbial landscape and antibiotic susceptibility of soft tissue infection pathogens among adults and children during the period 2018–2020. We studied 110 samples of pus admitted to the Scientific Research laboratory of the Karaganda Medical University from 2018 to 2020. Each sample was studied using the standard and express methods. The antibiotic susceptibility was determined by using the diffuse disk method in accordance with the CLSI 2018 recommendations. As such, 50% of *S. epidermidis* strains in children and 30% in adults were methicillin resistant. Differences in the resistance of *S. aureus* strains in children and adults were insignificant. Thus, methicillin-resistant *S. aureus* (MRSA) was not detected in children, but in adults, on the other hand, their percentage was 12.5%. The third cause of infection in adults was *E. coli* (13.72%), among which 75% were multidrug resistant. *A. baumanii* was found in 4.9% of adult patients’ samples, of which 60% were multidrug resistant. The effectiveness of the most prescribed antibiotics decreased due to the isolated strain resistance.

## 1. Introduction

Skin and soft tissue inflammatory diseases of bacterial origin occupy a significant part of calls to emergency departments. One of the most common types of infection is soft tissue infection. Therefore, this type of infection is in third place, with respect to the frequency, among all causes of sepsis. Further, 24% of nosocomial infections are due to soft tissue infections. In 2018, 271,282 patients were admitted to hospitals with a diagnosis of skin and soft tissue infections, which accounted for 1.5% of the total population, or 5237.3 people per 100,000 population. [1]. Among others, Nathwani et al. presented the evidence that 75% of working-age patients have at least one hospitalization for soft tissue infections. Outpatient treatment of such patients takes from 13.6 to 17.6 days; inpatient treatment leads to 18.5–23.8 days of labor loss [2].

Recently, there has been an increase in the incidence of this pathology among children. It has been suggested that this is due to an increase in the incidence of methicillin-resistant Staphilococci.

Gram-negative cocci, *Escherichia coli* and *Pseudomonas aeruginosa*, are the leading causes of soft tissue infections. In addition, it is noted that these types of infection are most often causative agents in mono- or polymicrobial infections [3,4,5,6,7,8].

Lim et al. had a similar point of view; they pointed out that due to the spread of methicillin-resistant strains of Staphylococci that are resistant to various groups of antibiotics used in the treatment of soft tissue infections, such as beta-lactams, the efficacy of that antibiotics decreases [5]. It is also noted that previously uncommon strains of Gram-negative pathogens with resistance to antibiotics tend to be more widespread [9,10,11,12,13]. Lim et al. noted a decrease in the effectiveness of treatments for skin and soft tissue infections due to the wide spread of methicillin-resistant strains. There is a tendency to change the etiologic agent in soft tissue infection from Gram + MRSA to Gram-negative flora, with distinct resistance profiles, and the etiological agent, in any case, has certain resistance mechanisms [9,10,11,12,13].

Thus, the dynamics of the microbial landscape and antibiotic susceptibility of pathogens in soft tissue infections among children and adults for the period 2018–2020 remains unclear, which was the purpose of this study. To do this, we studied the differences in the etiology of infection in these groups of patients. Determining the rational use of antibiotics in skin and soft tissue infections and the designation of high-quality infection control in the hospitals, both are very important tasks that need to be carried out on an ongoing basis and be carefully monitored.

## 2. Results

The study included patients who applied to hospitals in the city of Karaganda with soft tissue infections of various localization, both adults and children. The average age of children was 12 ± 7.4 years, one infant was 0.33 months; in the group of adult patients, the average age was 43.9 ± 17.9 years (Table 1).

Of the 110 samples, 100% were positive cultures. Samples were isolated from adults and children. Mixed cultures were present in 12.73% (14) of samples, two samples of which were samples taken from children (25%). There was no statistically significant relationship between the frequency of mixed isolates and the age of the participants (*p* < 0.05).

These data also confirm the results of other similar studies, according to which, soft tissue infections were most often represented by abscesses, phlegmons, boils and atheromas. According to other studies among children, the most common types of these infections are impetigo, skin abscess, and cellulitis/orbital cellulitis. Among adults, abscesses, phlegmons and atheromas are noted [4,9,10,14]. The described data correlate with the data obtained in our study. Thus, Gram-negative bacteria in children were detected statistically more often (*p* = 0.026) than Gram-positive ones, in a ratio of 5:3 (62.5%:37.5%). These data are presented in Table 2. The structure of pathogens in children and adults are presented in the Figure 1 and Figure 2 below.

*Staphilococcus epidermidis* (*S. epidermidis*) was present in 25% of samples taken from children, while *Staphilococcus aureus* (*S. aureus*) was present in 32.34% of adult samples. The ratio of Gram-positive to Gram-negative bacteria in adults was 53:49 (54.1%:45.9%). Slightly less than *S. aureus* in adults, the bacteria from the *Enterobacterales* family (*E. coli* (13.7%) and *Acinetobacter baumannii* (*A. baumanii*) (4.9%)) were also sown in the structure of pathogens (Table 3). In the first place, in the etiological structure of soft tissue infections in children, *S. epidermidis* was isolated (25%). *A. baumannii*, *Streptococcus beta-haemolytic*, *Enterobacter aerogenes*, *Enterococcus faecium*, *and Staphylococcus haemolyticus* were detected in equal shares of 12.50%. The data are consistent with Galli et al., who argued that this is due to a decrease in immunity and the fact that the causative agent is mainly one’s own skin [15]. The etiological structure of pathogens of soft tissue infections among adult patients is more diverse and pathogenic (Table 3). In 32.34% of cases, *S. aureus* was isolated, which is the dominant (predominant) microorganism [14,15,16,17]. The dynamics of the pathogens during 2018–2020 is presented on a Figure 3 below. 

For the period from 2018 to 2020, we established the following dynamics of the species composition of bacteria sown during soft tissue infections: the frequency of *S. aureus* isolation from 8 strains (22.2%) in 2018 increased to 16 (38.1%) in 2020; *E.coli* decreased from 19.4% in 2018 to 9.4% and 9.5% in 2019 and 2020, respectively. Most often % *E.coli*, *P. aeruginosa*, *S. aureus* and *S. epidermidis* were sown. For the period 2018–2020, the dynamics of *S. aureus* isolation can be traced. In 2018, *S. aureus* accounted for 22.2% (*n* = 8) of isolated strains, and by 2020, *S. aureus* was isolated in 38.1% (*n* = 16).

A comparison of the species composition in different years was carried out using the Z-criterion for pairs.

For *S. epidermidis*, statistically significant differences were found when comparing the number of strains in 2018 and 2019, and 2018 and 2020; for *P. aeruginosa*, when comparing the number of strains in 2018 and 2020.

When assessing the dynamics of the sensitivity of isolated strains of microorganisms to antibacterial drugs, it was found that for the period 2018–2020, *P. aeruginosa* remained highly sensitive to polymyxin (100%), increased sensitivity to fluoroquinolones (levofloxacin—from 50% in 2019 to 60% in 2020, ciprofloxacin—from 50% in 2019 to 66.6% in 2020); low sensitivity to cephalosporins is noted. Both strains isolated in 2019 were resistant to most drugs in this group: in 2020, sensitivity to ceftazidime was 50%, cefotaxime—40%, cefepime—60%. In 2020, there was an increase in the sensitivity of *P. aeruginosa* to aztreonam (from 50 to 83.3%) and meropenem (from 50% to 66.6%); sensitivity to imipenem, on the contrary, decreased from 100% to 66.6%. There was also a decrease in sensitivity to aminoglycosides: kanamycin and amikacin—from 100% to 66.6%; sensitivity to tobramycin increased from 50% to 66.6%.

*P. aeruginosa* isolates were sensitive to polymyxin (100%), aztreonam (83.3%), imipenem (66.6%), meropenem (66.6%), fluoroquinolones–norfloxacin (66.6%), ciprofloxacin (66.6%), ofloxacin (66.6%), and levofloxacin (60%). At the same time, one-third of the isolated strains were resistant to carbapenems and fluoroquinolones. In addition, the isolates were resistant to cephalosporins (cefotaxime (60%), ceftazidime (50%) and cefepime (40%), penicillins used in the treatment of *Pseudomonas aeruginosa* infection (carbenicillin—100%, azlocillin—100%, piperacillin—60%, piperacillin/tazobactam—60%, ticarcillin/sulbactam—33.3%. Further, 66.6% of *P. aeruginosa* strains were moderately sensitive to ticarcillin/sulbactam and 40% to piperacillin and piperacillin/tazobactam.

In second place, in terms of prevalence among Gram-negative flora, were strains of *E. coli*—4 cases (9.5%). When analyzing the sensitivity of *E. coli* strains, it was found that the isolates were sensitive in 63% of cases, resistant in 30% and moderately sensitive in 6.8% of cases. There is a preservation of high sensitivity to carbapenems (100%), polymyxin (100%), and chloramphenicol (100%); sensitivity to aminoglycosides, compared with 2018, decreased in 2019, to gentamicin (up to 66.6%) and tobramycin (up to 66.6%); in 2020, sensitivity to gentamicin was 50%, to amikacin—66.6%, netilmicin—66.6%, tobramycin—75%. Sensitivity to fluoroquinolons, on the contrary, increased from 50% in 2018 and 2019 (for ofloxacin) to 75% in 2020, and for ciprofloxacin, from 66.6% in 2018–2019, up to 75% in 2020. In 2019–2020 there was an increase in the resistance of *E. coli* strains to penicillins and cephalosporins. In 2019, three strains were isolated and all of them were resistant to ampicillin, ampicillin/sulbactam, amoxicillin, cefuroxime, cefotaxime, and cefepime. In 2020, the isolated strains were resistant, in 100% of cases, to ampicillin and cefuroxime, in 66.6% to cefotaxime, in 50% to cefepime and ampicillin-sulbactam, and in 33.3% to amoxicillin.

In samples obtained from both groups, an increasing trend towards resistance was found.

Strains of *S. epidermidis* isolated from children were resistant to the most-used antibiotics in one (50%) case (MRSE was identified). In adults, this figure was six (30%). All strains were susceptible to fusidic acid, vancomycin, and linezolid 20 (100%). When assessing the dynamics of sensitivity in *S. epidermidis* strains, a gradual decrease in sensitivity to cefoxitin is noted (from 88.8% in 2018 to 28.6% in 2020). All isolated strains retained sensitivity to vancomycin (100%) and fusidic acid (100%). In 2019, there was a decrease in sensitivity to tetracycline (from 100% in 2018 to 50%) and ciprofloxacin (from 100% in 2018 to 75%); however, in 2020, high sensitivity to these drugs was noted. The sensitivity of *S. epidermidis* strains to antibiotics is shown in Figure 4.

Vancomycin-resistant strains were not identified in our study. These data allow us to conclude that the antibiotic is highly effective for the treatment of soft tissue infections caused by Gram-positive bacteria.

*S. aureus* (32.34%) has one of the highest occurrence rates. The MRSA rate in adults was 12.5%. In 2018, all isolated strains remained highly sensitive to antibiotics; in 2019, one strain (10%), resistant to beta-lactams, fluoroquinolones and gentamicin, was isolated, and in 2020, five (31.3%) strains were resistant to three and more beta-lactam antibiotics and fluoroquinolones, of which one (6.3%) retained sensitivity only to vancomycin, linezolid and tetracycline.

## 3. Discussion

During the study and analysis of the results, it was found that, most often, *S. epidermidis* was isolated from children (25%); in adults, *S. epidermidis* (19.6%) and *S. aureus* (32.34%). Based on the obtained results, throughout 2020, the antibiotic resistance of pathogens increased (*p* = 0.026). The findings are also consistent with those of another study by Hu et al. [6]. Further, there is a tendency to increase the resistance of the following strains of bacteria isolated from the adult population: *S. epidermidis* (MRSE) and *S. aureus* (MRSA). For 2020, the values were 30% and 12.5%, respectively. In children, in the same year, this indicator for MRSE was 50%. Similar data were obtained in studies by other scientists [16,17,18]. According to the results of our study, it was found that *P. aeruginosa* strains are highly sensitive to polymyxin and fluoroquinolones and insensitive to cephalosporins; sensitivity to aminoglycosides in dynamics decreases. There is also an increase in the resistance of *E. coli* strains to penicillins, cephalosporins, and aminoglycosides.

The general trend towards an increase in the resistance of microorganisms is also noted by the studies of other scientists [4,6,8,19]. The negative trend in the growth of resistance to cefoxitin, azithromycin, oxacillin, and amikacin, including those classified as reserve according to the AWARE classification, is probably associated not only with the uncontrolled over-the-counter dispensing of antimicrobials, but also with the low accessibility of practitioners prescribing antibacterial drugs for information on both the data of microbial landscape dynamics and antimicrobial susceptibility among microbes. This can be indirectly judged by the results of other studies, which also pointed out reduced control over the sale of antibacterial drugs and their irrational prescription, as a key aspect that caused an increased trend toward bacterial resistance. Other reviewers also noted that developing countries have difficulties with the appropriate use of antibiotics and the lack of national and regional guidance on antibiotic use [11,20,21,22]. Just like in our research J-F. Jabbour et al. observed that in the case of soft tissue infections concerning multidrug-resistant strains, novel antibiotics have shown high efficiency [23,24,25]. This may be due to the low prevalence of these types of antibiotics, as well as their low prescription by doctors. In addition, there is likely to be low access to these antibiotics in developing countries. We also analyzed studies on bacterial resistance to antibiotics in other types of infections. Thus, there is a trend towards increased resistance to ciprofloxacin and methicillin in diabetic foot infections [26,27]. More detailed conclusions are difficult to draw due to the lack of relevant studies on this topic. The study was conducted as part of a master’s thesis and is also limited by the pandemic restrictions. In this regard, it was decided to limit the data only to the Karaganda region, and the amount of data for analysis was minimally possible but made up a representative sample of the general population [28]. The results from our study revealed the growth of antimicrobial resistance (AMR) and the need for continuous microbiological monitoring and timely reporting of emerging trends and dynamics, both at the level of hospitals and at the country level. To achieve this goal, the service of reference and sentinel laboratories is being strengthened in the Republic of Kazakhstan. Currently, there are five reference laboratories, in which modern methods and standards have been introduced (MALDI-TOF mass spectrometry, EUCAST recommendations and microbiology laboratory database software WHONET), and two more laboratories plan to launch in the near future The priority tasks, currently, are the expansion of sentinel laboratories in medical organizations and the introduction of unified standards for laboratory diagnostics, increasing the level of knowledge of microbiologists, which, in turn, is designed to ensure the involvement of all participants in the AMR containment process, to raise awareness of the rational use of antibiotics at all levels of health care. Methodical assistance of working groups (microbiologists, clinical pharmacologists, epidemiologists, infectious disease specialists) in the creation and functioning of local monitoring, as well as enhanced academic training of personnel, are necessary measures in the short term to control and constrain the AMR.

## 4. Materials and Methods

The study was conducted in the Scientific Research Laboratory at the Medical University of Karaganda (Kazakhstan). The material for testing consisted of 110 pediatric and adult patient’s samples from the Karaganda region diagnosed with soft tissue infections of different localizations and severity. Patients were divided into 2 groups by age: children from 0 to 18 years old and adults from 18 to 75 years old.

Bio sampling was carried out directly in the hospital from patients with aseptic technique into HI culture Transport Swabs Amies Medium (Himedia, Mumbai, India) before the beginning of antibiotic therapy. The transportation of biomaterial samples to the Scientific Research Laboratory of the Medical University of Karaganda was carried out within 2 h in compliance with the temperature and time regimes and safety standards when handling biological material. Plating was carried out on following media: blood agar, yolk-salt agar, Sabouraud’s medium. After isolation of a pure culture, the identification of microorganisms was carried out by the mass spectrometry method (Microflex–LT, Biotyper System, Bruker Daltonics, Bremen, Germany) [29].

Identification was considered successful at the level of species with a high degree of confidence when the score exceeded 2.0; if the score was between 2.0 and 1.7, identification was considered successful at the genus level with adequate confidence [30].

The determination of antimicrobial sensitivity was carried out for the microorganisms for each group of patients. Susceptibility of bacterial isolates to antibacterial drugs was carried out by the disk diffusion method to the following antibiotics: oxacillin (1 µg), cefoxitin (30 µg), ampicillin (10 µg), cefepime (30 µg), imipenem (10 µg), meropenem (10 µg), amikacin (30 µg), kanamycin (30 µg), streptomycin (10 µg), gentamycin (10 µg), netilmicin (30 µg), tobramycin (30 µg), nalidixic acid (30 µg), ciprofloxacin (5 µg), levofloxacin (5 µg), linezolid (30 µg), clindamycin (2 µg), rifampicin (5 µg), azithromycin (15 µg), fusidic acid (10 µg), and tetracycline (30 µg); colistin, vancomycin, teicoplanin MICs were determined in accordance with the CLSI M100 ED32:2022 recommendations. Analysis of the results was carried out following the recommendations of CLSI M100 ED32:2022 [31].

Phenotypic detection of methicillin-resistant *S. aureus* (MRSA) and *S. epidermidis* (MRSE) was conducted by cefoxitin disk (30 µg). MRSA-positive strains had a zone of inhibition <21 mm, and MRSE positive strains had a zone of inhibition <24 mm [31].

Internal quality control was carried out on the control strains: *Staphylococcus aureus* ATCC 25923, *Pseudomonas aeruginosa* ATCC 27853, *Escherichia coli* ATCC 25922 (ESBL and AmpC−), *Klebsiella pneumoniae* WHO-3 (ESBL+), and *Enterobacter cloacae* WHO-238 (AmpC+ and ESBL−).

The analysis of susceptibility to antibacterial drugs was carried out by calculating the 95% confidence interval using the WHONET 5.6 program. Statistical analysis was carried out in the STATISTICA 7.0 program using the Z-criterion and Student’s criterion; for quantitative indicators, descriptive statistics were calculated; for qualitative indicators, a frequency analysis was carried out. The distribution of indicators was carried out using the Shapiro–Wilk test. In each age group, the average values of age and standard deviation were calculated. The relative frequency of occurrence of a qualitative trait in each sample was calculated.

## 5. Conclusions

Nowadays, there is an increase in the resistance of microorganisms to antibiotics. In our study, it was determined that in the most frequently isolated strains of microorganisms (*S. epidermidis*, *S. aureus*, *E. coli*, *P. aeruginosa*) in the hospital, the sensitivity to antibiotics commonly used for soft tissue infections decreases over time. Based on this, the range of effective agents for the treatment of soft tissue infections in practitioners is becoming smaller. As a consequence, the preventing of pan-drug resistance is becoming ever more pronounced. Differences in resistance to most of the studied strains in children and adults were insignificant, which also causes concern. In this regard, it is necessary to take several measures, aimed at eliminating this threatening trend. We propose to amend national guidelines on the prudent use of antibiotics, as well as to raise public awareness of the prudent use of antibiotics, especially, to strengthen control over the use of antibiotics and rationalize antibiotic therapy. It is recommended to introduce, at the national level, the requirement for two-step microbiological analysis: before starting antibiotic therapy and at the end of therapy. Among other things, strong and effective measures should also be taken to increase the availability of up-to-date information on leading microorganisms and their antibiotic susceptibility to every doctor in the country.

## Figures and Tables

**Figure 1 antibiotics-11-00659-f001:**
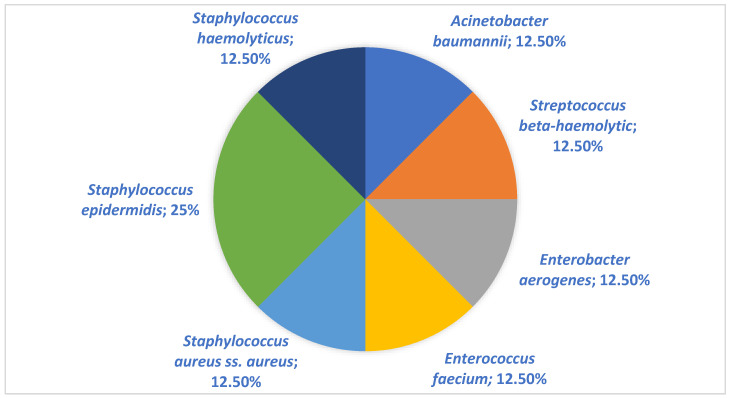
Etiology of soft tissue infections in Karaganda, Kazakhstan, 2018–2020, children.

**Figure 2 antibiotics-11-00659-f002:**
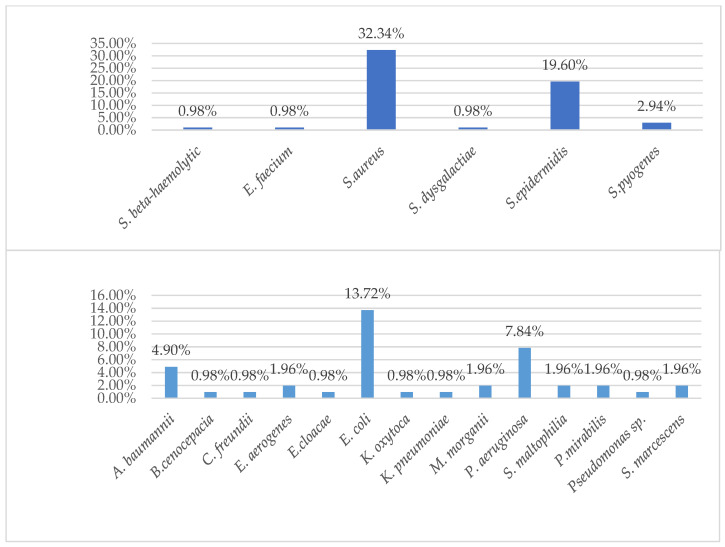
Etiology of soft tissue infections in Karaganda, Kazakhstan, 2018–2020, adults. Abbreviations: Gram-positive: *S. beta-haemolytic*—*Streptococcus beta-haemolyticus*, *E. faecium*—*Enterococcus faecium*, *S. aureus*—*Staphylococcus aureus ss. aureus*, *S. dysgalactiae*—*Streptococcus dysgalactiae ss. dysgalactiae*, *S. epidermidis*—*Staphylococcus epidermidis*, *S. pyogenes*—*Streptococcus pyogenes.* Gram-negative: *A. baumanii*—*Acinetobacter baumanii*, *B.cenocepacia*—*Burkholderia cenocepacia* (*genomovar III*), *C. freundii*—*Citrobacter freundii*, *E. aerogenes*—*Enterobacter aerogenes*, *E. cloacae*—*Enterobacter cloacae*, *E. coli*—*Escherichia coli*, *K. oxytoca*—*Klebsiella oxytoca*, *K. pneumoniae*—*Klebsiella pneumoniae ss. pneumoniae*, *M. morganii*—*Morganella morganii ss. morganii*, *P. aeruginosa*—*Pseudomonas aeruginosa*, *S. maltophilia*—*Stenotrophomonas maltophilia*, *P. mirabilis*—*Proteus mirabilis*, *S. marcescens*—*Serratia marcescens*.

**Figure 3 antibiotics-11-00659-f003:**
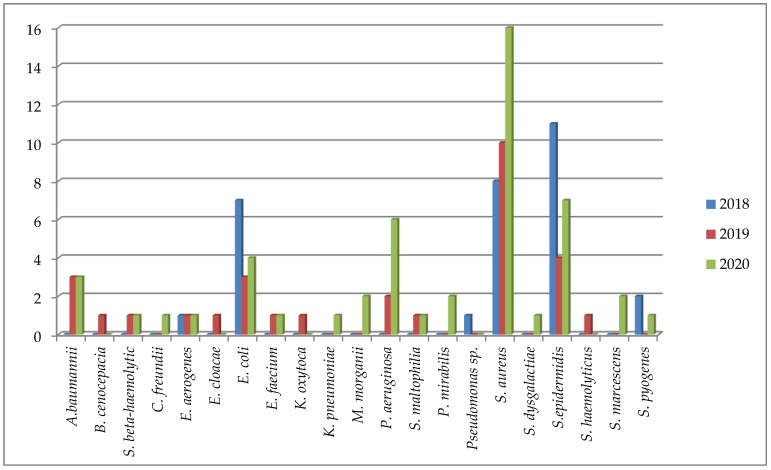
Soft tissue infections etiology structure in Karaganda, Kazakhstan, 2018–2020. Abbreviations: *A. baumanii*—*Acinetobacter baumanii*, *B. cenocepacia*—*Burkholderia cenocepacia* (*genomovar III*), *S. beta-haemolytic*—*Streptococcus beta-haemolytic*, *C. freundii*—*Citrobacter freundii*, *E. aerogenes*—*Enterobacter aerogenes*, *E. cloacae*—*Enterobacter cloacae*, *E. coli*—*Escherichia coli*, *E. faecium*—*Enterococcus faecium*, *K. oxytoca*—*Klebsiella oxytoca*, *K. pneumoniae*—*Klebsiella pneumoniae ss. pneumoniae*, *M. morganii*—*Morganella morganii ss. morganii*, *P. aeruginosa*—*Pseudomonas aeruginosa*, *S. maltophilia*—*Stenotrophomonas maltophilia*, *P. mirabilis*—*Proteus mirabilis*, *S. aureus*—*Staphylococcus aureus ss. aureus*, *S. dysgalactiae*—*Streptococcus dysgalactiae ss. dysgalactiae*, *S. epidermidis*—*Staphylococcus epidermidis*, *S. marcescens*—*Serratia marcescens*, *S. pyogenes*—*Streptococcus pyogenes*.

**Figure 4 antibiotics-11-00659-f004:**
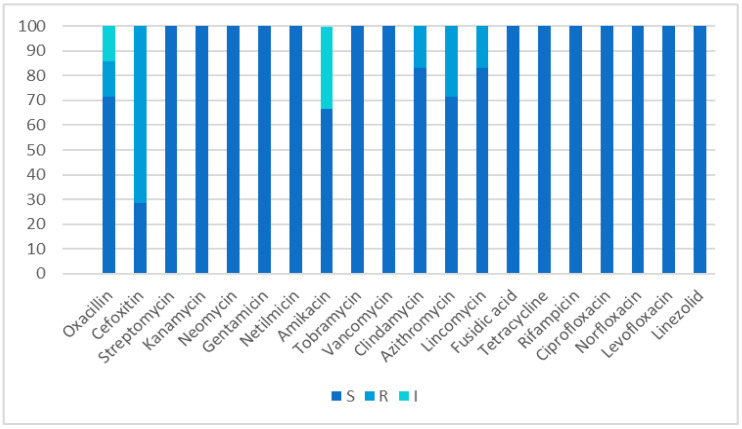
The susceptibility of *S. epidermidis* to antimicrobials in Karaganda, Kazakhstan, 2018–2020, %.

**Table 1 antibiotics-11-00659-t001:** Characterization of patients by age in Karaganda, Kazakhstan, 2018–2020.

Group of Patients	Age	Mean Age ± SD	Total Examined	Number of Positive Samples
Children	0–12 months	0.33 m	1	8 (100%)
	1–18 years	12 ± 7.4	7	
Adults	18–75 years	43.9 ± 17.9	102	102 (100%)

**Table 2 antibiotics-11-00659-t002:** Characterization of patients by diagnosis in Karaganda, Kazakhstan, 2018–2020.

Infection Localization	Children		Adults	
	Abs.	%	Abs.	%
Purulent wound discharge	2	25	20	19.60
Phlegmon	2	25	18	17.64
Abscesses	2	25	16	15.68
Boils	2	25	13	12.75
Atheroma			10	9.80
Postsurgery complications			8	7.84
Purulent appendicitis			4	3.92
Paraproctitis			4	3.92
Purulent complications in hematology			4	3.92
Erysepals			3	2.94
Purulent complications in cancer			2	1.76
**Total**	**8**	**100**	**102**	**100**

**Table 3 antibiotics-11-00659-t003:** Etiology of soft tissue infections in Karaganda, Kazakhstan, 2018–2020.

Pathogen	Children (*n* = 8) (%)	Adults (*n* = 102) (%)
*Acinetobacter baumannii*	1 (12.5%)	6 (5.88%)
*Burkholderia cenocepacia* (*genomovar III*)		1 (0.98%)
*Streptococcus beta-haemolytic*	1 (12.5%)	1 (0.98%)
*Enterobacter aerogenes*		2 (1.96%)
*Escherichia coli*		14 (13.72%)
*Enterococcus faecium*		2 (1.96%)
*Pseudomonas aeruginosa*	1 (12.5%)	8 (7.84%)
*Stenotrophomonas maltophilia*		2 (1.96%)
*Proteus mirabilis*	1 (12.5%)	2 (1.96%)
*Staphylococcus aureus ss. aureus*	1 (12.5%)	33 (32.34%)
*Staphylococcus epidermidis*	2 (25.0%)	21 (20.59%)
*Staphylococcus haemolyticus*	1 (12.5%)	1 (0.98%)
*Streptococcus pyogenes*		3 (2.94%)

## Data Availability

Database is available upon request to corresponding author.

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
