# Peer review of "Microbial Landscape and Antibiotic Susceptibility Dynamics of Skin and Soft Tissue Infections in Kazakhstan 2018–2020"

_antibiotics, 2022, doi:10.3390/antibiotics11050659_

Round 1

Reviewer 1 Report

Microbial landscape and antibiotic susceptibility dynamics of skin and soft tissue infections in Kazakhstan in 2018-2020

This is an interesting study that needs the authors to need to pay attention to the write-up to make to bring clarity to the aims.

Please clarify the main aim of the study

Line 39; namely, 271,292 cases of hospitalization of patients with this pathology occurred in 2018, not clear it supports the main reason on line 38 or it is a different statement.

Line 47 ‘gram-negative staphylococci” I will rather use “gram-negative cocci”

Line 53—the meaning of the statement is not clear

Line 66-68. Please rephrase the sentence

Line 75   ”Slightly less than S. aureus in adults, bacteria of the” this statement is not complete, what are you comparing s. aureus with

Line 76. I will suggest someone with English background go over the manuscript

What is the difference between table 2 and figure 1

LINE 129--“while this phenomenon was not found in children” not sure what phenomenon you are referring to.

I do not see the relevance of the work and the connection with the factors described in the discussion ( eg  175-178). The results are not well discussed and their relevance is not expressed

Materials and methods

How was the sample taken?

Line 205 can you please change Sowing to plating

Line 225, the confirmation of ESBL is not standard

Conclusion

The conclusion is not unique to your aims. Also, the tables in the manuscript are cluttered can you be kind and remove the lines from the tables

Author Response

Dear Reviewer!

Thank you very much for your comments and recommendations!

Please find out our answers for your comments below.

Point 1: Please clarify the main aim of the study

 Response 1:

The text “The aim of this study was to determine the differences in antibiotic sensitivity of bacterial agents of various origins among adults and children.”

Was changed to

“The aim of this study was to determine the differences in microbial landscape and antibiotic susceptibility of soft tissue infections pathogens among adults and children during 2018-2020 years.”

Point 2: Line 39; namely, 271,292 cases of hospitalization of patients with this pathology occurred in 2018, not clear it supports the main reason on line 38 or it is a different statement.

Response 2:

the cases of hospitalization and the frequency of soft tissue infections are both belongs to the justification of the actuality and importance of our study.

The sentence “According to an annual statistical report conducted in Kazakhstan, the main reason for patients to seek help from surgeons is infections of the skin and soft tissues, namely, 271,292 cases of hospitalization of patients with this pathology occurred in 2018 [1].”

has been transformed to “In 2018, 271,282 patients were admitted to hospitals with a diagnosis of skin and soft tissue infections, which accounted for 1.5% of the total population, or 5237.3 people per 100,000 population.”

Point 3: Line 47 ‘gram-negative staphylococci” I will rather use “gram-negative cocci”

 Response 3: corrected as suggested. Thank you for this correction.

 Point 4: Line 53—the meaning of the statement is not clear

 Response 4: this paragraph “Lim et al. have a similar point of view, they point out that due to the spread of methicillin-resistant strains of Staphylococci that are resistant to various groups of anti-biotics using in the treatment of soft tissue infections, such as beta-lactams, the effec-tiveness of the treatment of this group of infectious diseases is reduced [5]. It is also noted that previously uncommon strains of gram-negative pathogens with resistance to anti-biotics tend to be more widespread [9 – 13].

was transformed to “Lim et al. have a similar point of view, they point out that due to the spread of methicillin-resistant strains of Staphylococci that are resistant to various groups of antibiotics using in the treatment of soft tissue infections, such as beta-lactams, the efficacy of that antibiotics decreases  [5]. It is also noted that previously uncommon strains of gram-negative pathogens with resistance to antibiotics tend to be more widespread [9 – 13]. Lim et al. noted a decrease in the effectiveness of treatment of skin and soft tissue infections due to the wide spread of methicillin-resistant strains. There is a tendency to change the etiologic agent in soft tissue infection from Gram+ MRSA to Gram negative flora, with distinct resistance. [9 – 13].

 Point 5: Line 66-68. Please rephrase the sentence

 Response 5: done, the text “This data also confirms the results of other similar studies, in particular, according which soft tissue infections were most often represented by abscesses, phlegmons, boils and atheromas [4, 9, 10 14 ]. This data is presented in Table 1.” Has been deleted

 Point 6: Line 75   ”Slightly less than S. aureus in adults, bacteria of the” this statement is not complete, what are you comparing s. aureus with

 Response 6: the sentence “Slightly less than S. aureus in adults, bacteria of the Enterobacterales family (E. coli (13.7%) and A. baumanii (4.9%)) were also sown.

was changed to “Slightly less than S. aureus in adults, bacteria of the Enterobacterales family (E. coli (13.7%) and A. baumanii (4.9%)) were also sown in the structure of pathogens (Table 2). In the first place in the etiological structure of soft tissue infections in children, S. epidermidis was isolated (25%). Acinetobacter baumannii, Streptococcus beta-haemolytic, Enterobacter aerogenes, Enterococcus faecium, Staphylococcus haemolyticus were detected in equal shares of 12.50%. The data are consistent with Galli et al., who argue that this is due to a decrease in immunity and the fact that the causative agent is mainly one's own skin [15]. The etiological structure of pathogens of soft tissue infections among adult patients is more diverse and pathogenic (Table 2). In 32.34% of cases, S. aureus was isolated, which is the dominant (predominant) microorganism [14-17].

 Point 7: Line 76. I will suggest someone with English background go over the manuscript

 Response 7: Thank you for this recommendation! The manuscript was checked by the native-English speaking person.

 Point 8: What is the difference between table 2 and figure 1

 Response 8: the figures has been removed from the manuscript.

 Point 9: LINE 129--“while this phenomenon was not found in children” not sure what phenomenon you are referring to.

 Response 9: the sentence “The MRSA rate in adults was 12.5%, while this phenomenon was not found in children.”

has been shortened to “The MRSA rate in adults was 12.5%”.

 Point 10: I do not see the relevance of the work and the connection with the factors described in the discussion ( eg  175-178). The results are not well discussed and their relevance is not expressed

 Response 10: the information provided in the mentioned raws belongs to the currently acting measures of control and monitoring of antibiotics resistance spreading. Also, we have reviewed this part and re-write it.

 Point 11: Materials and methods

How was the sample taken?

 Response 11: this information was provided in lines 200-208, which now are Lines 386-394: “Bio sampling was carried out directly in the hospital from patients with aseptic technique into HI culture Transport Swabs Amies Medium (Himedia, India) before the beginning of antibiotic therapy. The transportation of biomaterial samples to the Scientific Research Laboratory of the Medical University of Karaganda was carried out within 2 hours in compliance with the temperature and time regimes and safety standards when handling biological material. Plating was carried out on media: blood agar, yolk-salt agar, Sabouraud's medium. After isolation of a pure culture, the identification of microorgan-isms was carried out by the mass spectrometry method (Microflex-LT, Biotyper System, Bruker Daltonics, Germany) [29].”.

 Point 12: Line 205 can you please change Sowing to plating

 Response 12: done.

 Point 13: Line 225, the confirmation of ESBL is not standard

 Response 13: Only phenotypical methods was performed in our study. Also, this study has been performed in 2018-2020, and the available approved standards and methods and equipment both was used.

 Point 14: Conclusion

The conclusion is not unique to your aims. Also, the tables in the manuscript are cluttered can you be kind and remove the lines from the tables

Response 14: thank you for this recommendation! The tables and conclusions were changed.

Reviewer 2 Report

The authors focused on a very important issue in this day and age. The increasing drug susceptibility among staphylococci is a significant difficulty both diagnostically and therapeutically. in many cases. The manuscript is structured correctly with all the required chapters. The literature is well chosen, answering the questions and theses posed by the authors, following the flow of thought.
The authors are also not without errors, which are marked in yellow in the manuscript file and justified with appropriate commentary. The main complaint is the use of outdated recommendations - the current ones are: CLSI 2022 M100 ED32:2022 and EUCAST Version 12.0, valid from 2022-01-01. The doses of antibiotics used for drug susceptibility testing given by the authors are also questionable. For the results to be comparable and more reliable, they should be interpreted according to CLSI and EUCAST recommendations simultaneously. However, the authors apply separately CLSI recommendations for interpreting the results of the disc-diffusion method and EUCAST recommendations for testing the MIC value for colistin.

Author Response

Thank you very much for your comments and recommendations!

Please find out our answers for your comments below.

Point 1: line 20 – sensitivity has corrected to susceptibility.

Point 2: the word “stable” is extra, deleted.

Point 3: the text has changed to suggested “among which 75% were multidrug-resistant”

 Point 4: in adults, on the other hand, their percentage was......... - corrected

 Point 5: multi-drug resistant - changed

 Point 6: page 2 “with respect to” –  has changed

 Point 7: presented the evidence – corrected

on both the data of microbial landscape dynamics and antimicrobial susceptibility among microbes - changed

 Point 8: corrected to working-age patients

 Point 9: the first time of pathogens spelling now full

 Point 10: cause changed to causes

 Point 11: Perhaps the notation "14/total number of strains" would be more readable? – this part has re-phrased

 Point 12: 2 samples - rephrased

 Point 13: it may be appropriate to add n, the total number of children and adults taken into account  in the respective table cells as in Table 2. – this part has changed, please see the latest version.

 Point 14: enterobacterales in italic – done

Point 15: E.coli in italic – done

Point 16: this microbial name appear in the article for the first time so it is useful to write the full generic – done

Points on the pages 4-5 –this part has changed

Point page 6 – this part of text has changed

Page 7 – fully re-phrased

there is no description of the axes on the graph, it is not clear what the numbers on the vertical axis refer to. there is no description of the axes on the graph, it is not clear what the numbers on the vertical axis refer to.

Answer: description has added

remove the word "figure", it is unnecessary – done

ESBL – deleted, as not corresponding.

Page 8 – observed – done, as the rest text has changed

Currently corrected

Abbreviations were expanded and clarifications for EUCAST and WHONET has added

Page 9 – re-phrased.

Re-CLSI 2022 – this study was performed in 2018-2020, so the previous version has used.

Pan-resistance – changed.

Materials and Methods:

The group of children sampled for this study is too small to make comparisons with adults and to make judgments about the effectiveness or lack thereof of treatment with the drugs studied.

Answer: the children – all the cases where parents gave the consent were included into this study, so all the admitted samples were considered. The little amount of children included was due to the quarantine restrictions.

But those samples and the results of their analysis were important for the picture of the landscape and susceptibility understanding both.

Please also note that the purpose of our study was to assess the entire landscape and the dynamics of susceptibility, not the comparison between children and adults, therefore, all available and legally obtained samples were analyzed.

The breakpoints values for cefoxitin against S. aureus are currently ≤ 21 mm and for S. epidermidis ≤ 24 mm, according to CLSI 2022

Answer: corrected.

Page 10:

Does this mean that only selected microorganisms were considered and the rest were rejected?

Answer: no, the determination was performed for each identified pathogen, just only significant points were described in this paper.

“Prevailed” – deleted

Why did the authors choose only this antibiotic to determine the MIC value and why were the EUCAST recommendations applied in this situation? Why were they not also used to interpret the other results?

Answer: thank you very much for this comment! We have deleted the EUCAST recommendations due to CLSI 2018 sufficiency.

CLSI quality control strain is for disk diffusion: S. aureus ATCC®25923

EUCAST Quality control: Staphylococcus aureus ATCC 29213.

Answer: ATCC®25923 has mentioned in the highlighted line

Reviewer 3 Report

The authors presented an interesting article about the prevalence of resistant microorganisms in samples taken from patients affected by skin and soft tissues infections.

The manuscript would need an extensive revision, preferentially by a mothertongue writer, to improve its readability and comprehension.

Moreover, we found some critical issues we think the author should address before resubmitting this paper.

INTRODUCTION

Lines 37-40

According to an annual statistical report conducted in Kazakhstan, the main reason for patients to seek help from surgeons is infections of the skin and soft tissues, namely, 271,292 cases of hospitalization of patients with this pathology occurred in 2018

  • Please, break down to simple phrases the previous sentence to make it clearer.

Lines 50-54

Lim et al. have a similar point of view, they point out that due to the spread of methicillin-resistant strains of Staphylococci that are resistant to various groups of antibiotics using in the treatment of soft tissue infections, such as beta-lactams, the effectiveness of the treatment of this group of infectious diseases is reduced [5]. It is also noted that previously uncommon strains of gram-negative pathogens with resistance to antibiotics tend to be more widespread [9 – 13].

  • Please, break down to simple phrases the previous sentences to make them clearer. Moreover, the author should clarify what they mean by “the effectiveness of the treatment of this group of infectious diseases is reduced”

Lines 56-59

Thus, the dynamics of the microbial landscape and antibiotic susceptibility of pathogens of soft tissue infections among children and adults for the period 2018-2020 remains unclear, which was the purpose of this study. To do this, we studied the differences in the etiology of infection in these groups of patients.

  • Please, write the aim of the study in a clearer way.

Overall, the introduction section is not informative enough about the matter of the study, and it is poorly written. Please, correct punctuation and rephrase some sentences which slow down the comprehension of the paper.

RESULTS

Lines 62-65

2 samples of which were samples taken from children (25%). There was no statistically significant relationship between the frequency of mixed isolates and the age of the participants (p<0.05).

  • The authors should not begin a sentence with a number. Please change “2” with “Two”.
  • Please, write the exact value of “p” resulting from the test applied. Moreover, if it is <0.05, the relationship is statistically significant. However, the p value in this case is doubtful, because of the sample size. The author should comment on this fact within the “discussion” section.

Lines 73-74

  1. epidermidis was present in 25% of samples taken from children, while S. aureus was present in 32.34% of adult samples.
  • Please express the number of samples in which epidermidis and S. aureus were found as “count (percentage)”. Moreover, why did the authors report about two different bacteria in two different populations? Please clarify.

Lines 74-75

The ratio of Gram-positive to Gram-negative bacteria in adults was 53:49 (54.1%:45.9%).

  • In this case, ratio is not of immediate comprehension. The authors should find a clearer way to convey this information. In our opinion, the best way is “count (percentage)”.

Table 3 is missing in the count, while table 4 is presented twice for two different tables. Please correct.

Table 3

Table 4: Comparison of the pathogens in Karaganda, Kazakhstan, 2018-2020

Count and percentages should be added in the table. Moreover, the best test in this case would be Fisher test, which would allow to test if the growth is significant between all groups, and then perform a post-hoc analysis for smaller groups. Please change the table accordingly.

Lines 121-123

Strains of S. epidermidis isolated from children were resistant in 50% of cases (MRSE). In adults, this figure was 30%. All strains were susceptible to fusidic acid, vancomycin and linezolid (100%). The sensitivity of S. epidermidis strains to antibiotics is shown in Figure 2.

  • epidermidis is often naturally methicillin resistant. We do not find this data suggestive of any problem. Moreover, S. epidermidis was found only in two out of eight samples taken in children. This means that only one isolate was resistant to methicillin.
  • Please, express the number of strains as “count (percentage)”. Disguising the data, only referring to them as percentages, does not make the result more relevant.

Line 129

while this phenomenon was not found in children.

  • Resistance is not a phenomenon.

DISCUSSION

Lines 137-139

In addition, gram-negative bacteria in 137 children were detected statistically more often (p < 0.05) than gram-positive ones in a ratio 138 of 5:3 (62.5% : 37.5%).

  • This result was not presented within the “results” section. Please, refrain from commenting in the discussion results that were not presented. Moreover, we recommend again to show all the exact values of p.

MATERIALS AND METHODS

Lines 193-195 and Table 4

The average age of 193 children was 12 ± 7.4 years, 1 infant was 0.33m; in the group of adult patients, the average 194 age was 43.9 ± 17.9 years (Table 4).

  • This part should be presented in the results section. Moreover, did the authors test the distribution of the variable “age”? If not normally distributed, the variable should be more correctly expressed as “median (IQR)”.

Line 205

Sowing

  • This should be changed with the word “cultures”

Lines 232-233

Statistical analysis was carried out in the STATISTICA 7.0 program using the Z-criterion and Student’s criterion.

  • This is not sufficient to describe the statistical methods used to carry out the study.
  • Which variables were collected and how? Did the authors tested the normality of the variables before choosing any test to study significant relations? Which tests were used to study possible relations between variables? Please clarify.

Author Response

Dear Reviewer!

Thank you very much for your comments and recommendations!

Please find out our answers for your comments below.

Point 1: The manuscript would need an extensive revision, preferentially by a mothertongue writer, to improve its readability and comprehension.

Response 1: thank you very much for this recommendation! Now the manuscript has been rexised by native-English speaking person.

Point 2: INTRODUCTION

Lines 37-40

According to an annual statistical report conducted in Kazakhstan, the main reason for patients to seek help from surgeons is infections of the skin and soft tissues, namely, 271,292 cases of hospitalization of patients with this pathology occurred in 2018

Please, break down to simple phrases the previous sentence to make it clearer.

Response 2: This sentence has been transformed to “In 2018, 271,282 patients were admitted to hospitals with a diagnosis of skin and soft tissue infections, which accounted for 1.5% of the total population, or 5237.3 people per 100,000 population.”.

Point 3: Lines 50-54

Lim et al. have a similar point of view, they point out that due to the spread of methicillin-resistant strains of Staphylococci that are resistant to various groups of antibiotics using in the treatment of soft tissue infections, such as beta-lactams, the effectiveness of the treatment of this group of infectious diseases is reduced [5]. It is also noted that previously uncommon strains of gram-negative pathogens with resistance to antibiotics tend to be more widespread [9 – 13].

Please, break down to simple phrases the previous sentences to make them clearer. Moreover, the author should clarify what they mean by “the effectiveness of the treatment of this group of infectious diseases is reduced”

Response 3: corrected: “Lim et al. have a similar point of view, they point out that due to the spread of methicillin-resistant strains of Staphylococci that are resistant to various groups of antibiotics using in the treatment of soft tissue infections, such as beta-lactams, the efficacy of that antibiotics decreases  [5]. It is also noted that previously uncommon strains of gram-negative pathogens with resistance to antibiotics tend to be more widespread [9 – 13]. Lim et al. noted a decrease in the effectiveness of treatment of skin and soft tissue infections due to the wide spread of methicillin-resistant strains. There is a tendency to change the etiologic agent in soft tissue infection from Gram+ MRSA to Gram negative flora, with distinct resistance”.

 Point 4: Lines 56-59

Thus, the dynamics of the microbial landscape and antibiotic susceptibility of pathogens of soft tissue infections among children and adults for the period 2018-2020 remains unclear, which was the purpose of this study. To do this, we studied the differences in the etiology of infection in these groups of patients.

Please, write the aim of the study in a clearer way.

Overall, the introduction section is not informative enough about the matter of the study, and it is poorly written. Please, correct punctuation and rephrase some sentences which slow down the comprehension of the paper.

Response 4: the paragraph is corrected: “Thus, the dynamics of the microbial landscape and antibiotic susceptibility of pathogens of soft tissue infections among children and adults for the period 2018-2020 remains unclear, which was the purpose of this study. To do this, we studied the dif-ferences in the etiology of infection in these groups of patients. Determining the rational use of antibiotics in skin and soft tissue infections and designation of high-quality infection control in the hospitals both is very important task that needs to be carried out on an ongoing basis and carefully monitored.

 Point 5: RESULTS

Lines 62-65

2 samples of which were samples taken from children (25%). There was no statistically significant relationship between the frequency of mixed isolates and the age of the participants (p<0.05).

The authors should not begin a sentence with a number. Please change “2” with “Two”.

Please, write the exact value of “p” resulting from the test applied. Moreover, if it is <0.05, the relationship is statistically significant. However, the p value in this case is doubtful, because of the sample size. The author should comment on this fact within the “discussion” section.

Response 5: done: included into discussion

 Point 6: Lines 73-74

epidermidis was present in 25% of samples taken from children, while S. aureus was present in 32.34% of adult samples.

Please express the number of samples in which epidermidis and S. aureus were found as “count (percentage)”. Moreover, why did the authors report about two different bacteria in two different populations? Please clarify.

Response 6: the sentence was corrected: “Staphilococcus epidermidis (S. epidermidis) was present in 25% of samples taken from children, while Staphilococcus aureus (S. aureus) was present in 32.34% of adult samples. The ratio of Gram-positive to Gram-negative bacteria in adults was 53:49 (54.1%:45.9%). Slightly less than S. aureus in adults, bacteria of the Enterobacterales family (E. coli (13.7%) and Acinetobacter baumannii (A. baumanii)(4.9%)) were also sown in the structure of pathogens (Table 3). In the first place in the etiological structure of soft tissue infections in children, S. epidermidis was isolated (25%). A. baumannii, Streptococcus beta-haemolytic, Enterobacter aerogenes, Enterococcus faecium, Staphylococcus haemolyticus were detected in equal shares of 12.50%. The data are consistent with Galli et al., who argue that this is due to a decrease in immunity and the fact that the causative agent is mainly one's own skin [15]. The etiological structure of pathogens of soft tissue infections among adult patients is more diverse and pathogenic (Table 3). In 32.34% of cases, S. aureus was isolated, which is the dominant (predominant) microorganism [14-17]”.

 Point 7: Lines 74-75

The ratio of Gram-positive to Gram-negative bacteria in adults was 53:49 (54.1%:45.9%).

In this case, ratio is not of immediate comprehension. The authors should find a clearer way to convey this information. In our opinion, the best way is “count (percentage)”.

 Response 7: Thank you for this recommendation! This paragraph has been changed, the new text provided above.

 Point 8: Table 3 is missing in the count, while table 4 is presented twice for two different tables. Please correct.

 Response 8: the Tables numbering has been corrected.

 Point 9: Table 4: Comparison of the pathogens in Karaganda, Kazakhstan, 2018-2020

Count and percentages should be added in the table. Moreover, the best test in this case would be Fisher test, which would allow to test if the growth is significant between all groups, and then perform a post-hoc analysis for smaller groups. Please change the table accordingly.

Response 9: this part of the paper has been changed and the table has deleted

 Point 10: Lines 121-123

Strains of S. epidermidis isolated from children were resistant in 50% of cases (MRSE). In adults, this figure was 30%. All strains were susceptible to fusidic acid, vancomycin and linezolid (100%). The sensitivity of S. epidermidis strains to antibiotics is shown in Figure 2.

epidermidis is often naturally methicillin resistant. We do not find this data suggestive of any problem. Moreover, S. epidermidis was found only in two out of eight samples taken in children. This means that only one isolate was resistant to methicillin.

Please, express the number of strains as “count (percentage)”. Disguising the data, only referring to them as percentages, does not make the result more relevant.

 Response 10: this paragraph has corrected to corrected to: “Strains of S. epidermidis isolated from children were resistant to mostly used anti-biotics in 1 (50%) of cases (MRSE). In adults, this figure was 6 (30%). All strains were susceptible to fusidic acid, vancomycin and linezolid 20 (100%). When assessing the dynamics of sensitivity of S. epidermidis strains, a gradual decrease in sensitivity to cefoxitin is noted (from 88.8% in 2018 to 28.6% in 2020). All isolated strains retained sensitivity to vancomycin (100%) and fusidic acid (100%). In 2019, there was a decrease in sensitivity to tetracycline (from 100% in 2018 to 50%) and ciprofloxacin (from 100% in 2018 to 75%), however, in 2020, high sensitivity to these drugs is still noted. The sensitivity of S. epidermidis strains to antibiotics is shown in Figure 4.”.

 Point 11: Line 129

while this phenomenon was not found in children.

Resistance is not a phenomenon.

 Response 11: done, rephrased.

 Point 12: DISCUSSION

Lines 137-139

In addition, gram-negative bacteria in children were detected statistically more often (p < 0.05) than gram-positive ones in a ratio 138 of 5:3 (62.5% : 37.5%).

This result was not presented within the “results” section. Please, refrain from commenting in the discussion results that were not presented. Moreover, we recommend again to show all the exact values of p.

Response 12: done, rephrased with data.

 Point 13: MATERIALS AND METHODS

Lines 193-195 and Table 4

The average age of 193 children was 12 ± 7.4 years, 1 infant was 0.33m; in the group of adult patients, the average 194 age was 43.9 ± 17.9 years (Table 4).

This part should be presented in the results section. Moreover, did the authors test the distribution of the variable “age”? If not normally distributed, the variable should be more correctly expressed as “median (IQR)”.

Response 13: this part has changed: the table was deleted, the paragraph inserted into the Results section: “The average age of children was 12 ± 7.4 years, 1 infant was 0.33m; in the group of adult patients, the average age was 43.9 ± 17.9 years (Table 1).

Table 1: Characterization of patients by age in Karaganda, Kazakhstan, 2018-2020

Group of patients

Age

Mean age ± SD

Total examined

Number of positive samples

Children

0-12 months

0.33m

1

8 (100%)

1-18 years

12 ± 7,4

7

Adults

18-75 years

43,9 ± 17.9

102

102 (100%)

”.

 Point 14: Line 205

Sowing

This should be changed with the word “cultures”

Response 14: the phrase has changed.

 Point 15: Lines 232-233

Statistical analysis was carried out in the STATISTICA 7.0 program using the Z-criterion and Student’s criterion.

This is not sufficient to describe the statistical methods used to carry out the study.

Which variables were collected and how? Did the authors tested the normality of the variables before choosing any test to study significant relations? Which tests were used to study possible relations between variables? Please clarify.

Response 15: For quantitative indicators, descriptive statistics were calculated; for qualitative indicators, a frequency analysis was carried out. The distribution of indicators was carried out using the Shapiro-Wilco test.

Round 2

Reviewer 2 Report

The authors applied most of the suggested changes. However, there is still a lot work to do. The authors did not address the drug dosage information used in the drug susceptibility testing. An earlier version included the picogram unit, but it is still unclear for what reason. Besides, changing the notation of picogram to microgram (should be: μg) is insufficient in this situation in my opinion.

In addition, the authors took into account my suggestions for explanations under individual figures, but there is no notation of "Abbreviations" and the whole thing is not very readable. In my opinion, microbial abbreviations, division into Gram(+) and Gram(-) microorganisms, should be used to make the diagrams more legible and clear. The authors used italics in the names of microorganisms piecemeal, randomly, while the same notation should be found at least in the Figure 1.

And also, outdated versions of the EUCAST and CLSI recommendations still appear in the literature.

Reviewer 3 Report

The authors applied most of the suggested changes.

However, some changes are still required:

- The text still needs extensive English language revision.

- p values should be precisely indicated (i.e. p = 0.034, not p < 0.05)

- Frequencies should be expressed as "count (percentage)"

Response 15:

For quantitative indicators, descriptive statistics were calculated; for qualitative indicators, a frequency analysis was carried out. The distribution of indicators was carried out using the Shapiro-Wilco test.

The name of the test is Shapiro-Wilk.

Which descriptive statistics were used to summarize quantitative variables (not indicators)? Which frequency analyses were carried out for qualitative variables?

Please specify.
